# Neural cell integration into 3D bioprinted skeletal muscle constructs accelerates restoration of muscle function

Ji Hyun Kim[1], Ickhee Kim[1], Young-Joon Seol [1], In Kap Ko [1], James J. Yoo[1], Anthony Atala[1] & Sang Jin Lee [1✉]

A bioengineered skeletal muscle construct that mimics structural and functional characteristics of native skeletal muscle is a promising therapeutic option to treat extensive muscle defect injuries. We previously showed that bioprinted human skeletal muscle constructs were able to form multi-layered bundles with aligned myofibers. In this study, we investigate the effects of neural cell integration into the bioprinted skeletal muscle construct to accelerate functional muscle regeneration in vivo. Neural input into this bioprinted skeletal muscle construct shows the improvement of myofiber formation, long-term survival, and neuromuscular junction formation in vitro. More importantly, the bioprinted constructs with neural cell integration facilitate rapid innervation and mature into organized muscle tissue that restores normal muscle weight and function in a rodent model of muscle defect injury. These results suggest that the 3D bioprinted human neural-skeletal muscle constructs can be rapidly integrated with the host neural network, resulting in accelerated muscle function restoration.

[1] Wake Forest Institute for Regenerative Medicine, Wake Forest School of Medicine, Winston-Salem, NC 27157, USA. ✉email: sjlee@wakehealth.edu

Treatment of extensive muscle defect injuries due to trauma or tumor ablation poses a significant clinical challenge[1–4]. A muscle defect involving loss of >20% of the original mass invariably results in functional impairment with limited regeneration capacity, which requires reconstructive surgical procedures such as autologous muscle flap transfers[1]. However, limited availability of autologous muscle grafts and donor site morbidity often complicate these procedures[5]. Therefore, bioengineering of implantable skeletal muscle constructs that restore normal function of skeletal muscle would represent a significant advance in repairing extensive muscle defect injuries[6,7].

Tissue engineering strategies have focused on recapitulating the structural organization of native skeletal muscle, especially, uniaxial alignments of muscle cells, which is essential for the contractile properties of skeletal muscle and effective force generation for movement[8,9]. For example, mechanical stimulation through stretch-relaxation[10,11], anchors[12–14], and electrical stimulation[15,16] have been applied to induce the cellular alignment in bioengineered skeletal muscle constructs. Others have sought to control the cellular orientation directly using micro-patterned scaffolds[16–19]. These approaches have achieved bioengineered skeletal muscle tissue fabrication with cellular alignment in vitro, and some showed a degree of therapeutic potential in vivo. Also, a vascularization strategy using co-culture with endothelial cells (ECs) and fibroblasts improved the survival of the bioengineered skeletal muscle tissues[20,21]. This strategy has been applied to develop a three-dimensional (3D) human muscle model with endothelium specificity and endomysium for the study of fibrosis[22]. Importantly, the motor neurons have been cultured with the bioengineered muscle to better understand the development and maturation of neuromuscular junctions (NMJs)[23,24].

Currently, 3D bioprinting technologies have emerged as a powerful tool to build bioengineered skeletal muscle constructs[2,25–27], because these technologies can generate structurally complex cell-based constructs by precise positioning of multiple cell types, bioactive factors, and biomaterials within a single architecture to mimic native tissues[2,25,28]. We previously developed an implantable and biomimetic human skeletal muscle construct using 3D bioprinting strategies[2,25]. These constructs (mm$^3$–cm$^3$ scale) consisted of tens to hundreds of long parallel myofiber bundles, containing densely packed, highly viable, and aligned muscle cells. We also demonstrated the feasibility of using these 3D bioprinted human skeletal muscle constructs to treat critical-sized muscle defect injuries with structural and functional restoration in a rodent model[2]. While host nerve integration and formation of NMJs were evident within the implanted skeletal muscle constructs, we observed that the constructs did not support the full restoration of defected muscles at 8-week post-implantation, which is likely due to the delayed integration of host nerve.

Effective nerve integration of bioengineered skeletal muscle tissues remains a challenge for the reconstruction of extensively damaged muscle with the restoration of muscle function. Although current approaches to bioengineer skeletal muscle constructs have shown promising outcomes in vitro, only a few have validated the possibility of construct innervation with functional improvements in vivo[2,29]. Native skeletal muscle tissue is innervated through the peripheral nerve system by establishing NMJs, which are responsible for skeletal muscle survival, development, maturation, and contraction[30–32]. Denervated skeletal muscles lose contractility and undergo muscle atrophy[30,33]. Bioengineered skeletal muscle constructs consisting of cultured muscle cells that are similarly denervated require rapid integration with the host nervous system[33,34]. Otherwise, muscle atrophy and failure of functional recovery will occur.

To address the issue associated with delayed innervation, we develop human skeletal muscle constructs with neural cell integration by bioprinting human muscle progenitor cells (hMPCs) and human neural stem cells (hNSCs). We hypothesize that cellular interactions between hMPCs and hNSCs can promote muscle maturation and development and facilitate rapid integration with host nerve tissues in vivo. In addition, neural integration by pre-forming NMJs on myofibers within the construct would contribute to an increase in long-term survival and maturation of the bioengineered skeletal muscle construct before complete construct innervation occurs. It has been shown that neurotrophic factors (or neurotrophins) and neurotransmitters released from neural components and pre-formed NMJs increase skeletal muscle cell survival and differentiation[29,35–39].

In this study, we investigate the feasibility of using the bioprinted neural cell-integrated human skeletal muscle constructs to improve the structural and functional recovery of the extensive muscle defect injuries. The present study focuses on demonstrating the effects of neural cells and pre-formed NMJs on the bioprinted skeletal muscle constructs on muscle development, rapid innervation, and functional improvement. We evaluate the effect of neural cells on muscle cell viability, proliferation, and differentiation. Myogenic and neuronal differentiation and ability to form NMJs in the bioprinted constructs are also evaluated in vitro. To determine the feasibility of using this bioprinting approach to repair critical-sized muscle defect injuries, we apply the bioprinted constructs in a rat model of tibialis anterior (TA) muscle defect injury and evaluate the functional outcomes of muscle tissue reconstruction and innervation.

## Results

**Effects of neural cells on myotube formation.** To examine the effects of neural cells on skeletal muscle differentiation, hMPCs and hNSCs were co-cultured. We first optimized the ratio of skeletal muscle cells to neural cells in the two-dimensional (2D) co-culture system for the further 3D bioprinting experiment. As a high neural cell component to skeletal muscle cells is known to decrease myotube formations by interrupting the cell fusion[40], it is essential to optimize the ratio of cells to facilitate skeletal muscle cell maturation and tissue development, as well as NMJ formation. Myotube formation and long-term maintenance were evaluated in different ratios of hMPCs and hNSCs, ranging from 1:0 to 500:1 at 5 and 10 days in the differentiation medium. Immunofluorescence for myosin heavy chain (MHC) was performed, and the area of MHC$^+$ with different ratios of hMPCs and hNSCs was normalized to that of hMPC only (1:0). At 5 days of differentiation, the 100:1 and 300:1 ratio showed significantly increased myotube formation based on the area of MHC$^+$ compared to the hMPC only (Supplementary Fig. 1a, b). At 10 days, only the 300:1 ratio maintained the myotube formation compared to others (Supplementary Fig. 1c, d). In the 300:1 ratio, a higher number of myoD$^+$ and myogenin$^+$ cells were detected in the MPC + NSC group (Supplementary Fig. 2). In addition, when neural cell ratios were higher than 300:1, the presence of neural cells (higher than 300:1) seemed to be prevented the cell fusion, resulting in suppressed myotube formation. These results indicated that the optimal ratio of hMPCs and hNSCs was 300:1 on the myotube formation and long-term maintenance.

In the 300:1 ratio, the diameter and length of myotubes, the number of nuclei per myotubes, and striated myotubes all were increased when compared with the hMPC only (Fig. 1a–e). Glial and neuronal differentiation of hNSCs were also determined by immunofluorescence for glial fibrillary acidic protein (GFAP) and beta-III tubulin (βIIIT), and neurofilament (NF) (Fig. 1f). More importantly, innervation potential was confirmed by NMJ formation (white arrows, βIIIT$^+$/acetylcholine receptor (AChR)$^+$/MHC$^+$ and NF$^+$/AChR$^+$/MHC$^+$ NMJs) (Fig. 1g and

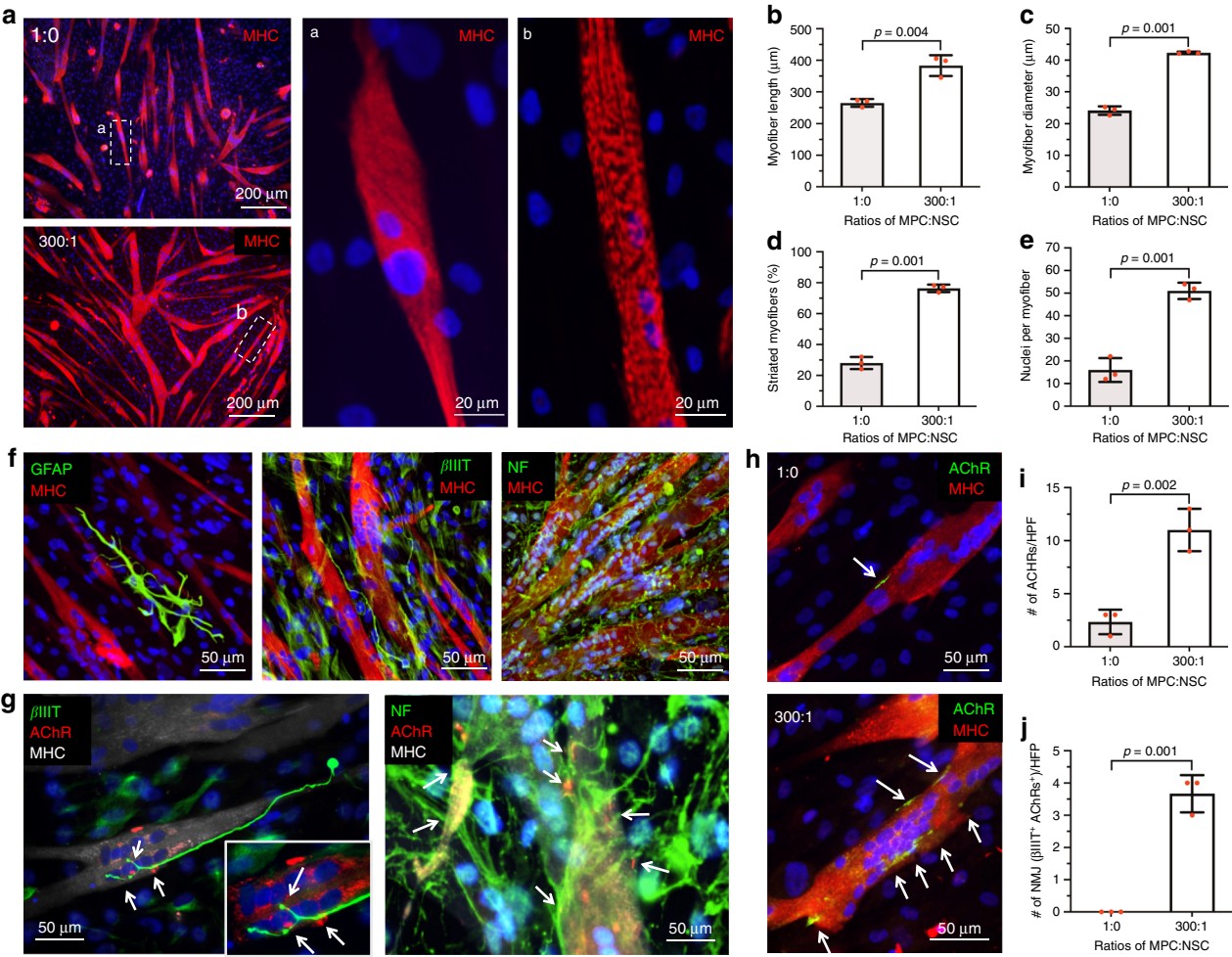

**Fig. 1 Two-dimensional (2D) co-culture of human muscle progenitor cells (hMPCs) and human neural stem cells (hNSCs) with different ratios.**
**a** Immunofluorescence for myosin heavy chain (MHC) (red)/DAPI (blue) of hMPCs culture (1:0) and co-culture (hMPCs:hNSCs = 300:1) at 5 days of differentiation; Striated myofiber structure of **a** hMPCs (1:0) and **b** co-culture of hMPCs and hNSCs (300:1). **b**–**e** Quantification of myofiber formation; **b** myofiber length (μm) ($n = 3$ per group), **c** myofiber diameter (μm) ($n = 3$ per group), **d** striated myofibers (%) ($n = 3$ per group), and **e** nuclei per myofiber ($n = 3$ per group). Co-culture group showed increased density, length, diameter, and striation of myofibers, as well as a number of nuclei per myofiber compared with the hMPCs only group. **f** Glial and neuronal differentiation of hNSCs in 2D co-culture. Immunofluorescence for glial fibrillary acidic protein (GFAP, glial cells, green)/MHC (red), beta-III tubulin (βIIIT, neurons, green)/MHC (red), and neurofilaments (NF, green)/MHC (red). **g**, **h** Acetylcholine receptor (AChR) clustering and neuromuscular junction (NMJ) formation (white arrows). **g** Immunofluorescence for βIIIT (green)/AChR (red)/MHC (gray) and NF(green)/AChR (red)/MHC (gray) in 2D co-culture. **h** Immunofluorescence for AChRs (green)/MHC (red) in hMPCs cultures and 2D co-culture. **i** The number of AChRs per field ($n = 3$ per group) and **j** co-localizations (βIIIT$^+$ AChRs per fields) ($n = 3$ per group). All data are represented as mean ± SD. The $p$-values by two-sided Student $t$-test are indicated.

Supplementary Fig. 3a). In the 300:1 ratio, increased AChRs expression on the myotubes and co-localizations with neurons were observed compared with hMPC only (white arrows, AChR$^+$/MHC$^+$ myotubes) (Fig. 1i, j). These results indicated that the introduction of neural cell components in skeletal muscle cell culture improved the differentiation and long-term maintenance of myotubes, neuronal differentiation of hNSCs, and NMJ formation.

**Muscle development of 3D bioprinted constructs in vitro.**
Based on the 2D cell culture experiment, the 300:1 ratio of hMPCs and hNSCs was used for bioprinting 3D skeletal muscle constructs (Fig. 2 and Fig. 3a, b). The bioprinted skeletal muscle constructs were evaluated by measuring cell viability, myotube formation ability, and NMJ formation. The live/dead staining assay of the bioprinted constructs showed higher cell viability in the MPC + NSC group (94.99 ± 0.74%) (mean ± SD) than the

MPC only group (85 ± 2.20%) at 1 day after printing (Fig. 3c, d), and the cells in the bioprinted constructs remained viable for the 7-day culture period.

MHC$^+$ myotube density (%) in the MPC + NSC group showed a 1.71-fold increase, and the formation of myotubes was evident by the presence of increased length of MHC$^+$ myotubes when compared with the MPC only group (Fig. 3e–g). In addition, the hNSCs in the bioprinted constructs were differentiated into GFAP$^+$ glial cells and βIIIT$^+$ neurons (Fig. 3h, i), and the βIIIT$^+$ neurites were found in contact with AChR clusters (NMJs) patterned on the myotubes in the printed constructs (Fig. 3j, k and Supplementary Fig. 3b). In the MPC only group, a few AChR clusters were found on the myotubes (Fig. 3l). Quantitatively, the numbers of AChRs per HPF (Fig. 3m) and NMJs (βIIIT+/AChR+) per HPF (Fig. 3n) were counted. The functionality of the NMJs in terms of synaptic transmission and calcium channels opening was determined by the calcium uptake imaging. We observed the increased number of cells with a high level of

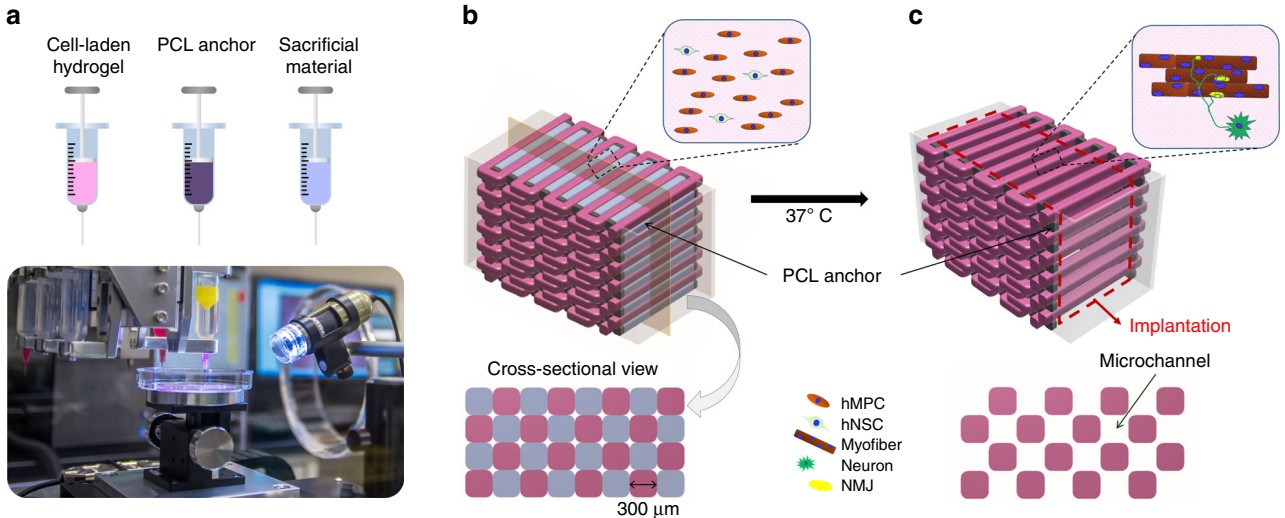

**Fig. 2 Bioprinting of human skeletal muscle constructs. a** Three components (cell-laden bioink, polycaprolactone (PCL), and sacrificial bioink) were printed in a layer-by-layer fashion using our custom 3D bioprinting system. **b** Design concept using 3D CAD modeling of the bioprinted construct. The cell-laden bioink containing hMPCs and/or hNSCs, the acellular sacrificing bioink, and the supporting PCL pillar were deposited by the multi-dispensing modules. **c** Microchannels in the constructs created after the removal of the sacrificial patterns to maintain the viability of printed cells.

intracellular calcium in the MPC + NSC group compared with the MPC only group (Supplementary Fig. 4). These results indicated that the bioprinted skeletal muscle constructs with hNSCs (MPC + NSC group) showed an increase in cell survival, muscle differentiation and maturation, and innervation potential based on AChRs pre-patterning and NMJs formation, compared with the MPC only group.

**Functional restoration in a muscle defect injury model.** Our rodent model of extensive muscle defect injury was created by removing 40% of the TA muscle mass followed by ablation of extensor digitorium longus (EDL) and extensor hallucis longus (EHL) muscles; these injuries caused irreversible anatomical and functional deformity following 6 months post-injury[2,41,42]. After creating the TA muscle defect, the bioprinted skeletal muscle constructs of MPC only and MPC + NSC were implanted into the defect sites for anatomical and functional skeletal muscle regeneration. Sham (without defect) and non-treated (defect only) groups were used as controls for comparison (a total of 24 immunodeficient RNU rats, $n = 3$ per group at the designated time points).

Surgically created muscle defects (non-treated group) did not show any recovery in the harvested TA muscle and resulted in severe muscular atrophy (Supplementary Fig. 5). In contrast, the bioprinted muscle constructs in the MPC only and MPC + NSC groups restored nearly their original TA muscle volume at 8 weeks post-implantation. The weight of TA muscle also improved in the bioprinted construct groups (both MPC only and MPC + NSC), as compared with the non-treated group, in which the values were not significantly different from the sham control at 8 weeks (Fig. 4a). Interestingly, the muscle weight in the MPC + NSC group was rapidly recovered at 4 weeks of implantation.

At 4 and 8 weeks post-implantation, the tetanic muscle force of the injured leg in response to electrical stimulation of peroneal nerve was evaluated through the in vivo functional analysis. Based on the muscle force measurement, the MPC + NSC group showed full restoration of muscle force when compared with the sham control at 8 weeks of implantation (100.82 ± 10.62 N·mm per Kg in sham group and 97.79 ± 14.44 N·mm per Kg in MPC + NSC group, *$p = 0.980$ between sham and MPC + NSC), while the MPC + NSC group was significantly higher than the MPC only

group (Fig. 4b). The MPC only group had 71.42% restoration of muscle force as compared with the sham group at 8 weeks after implantation. Thus, our results indicate that the introduction of neural cell components in the bioprinted skeletal muscle constructs could accelerate the muscle function restoration with extensive muscle defect injuries.

**Skeletal muscle regeneration.** We evaluated the newly formed myofibers, neural integration, and vascularization in the bio-printed skeletal muscle constructs at 4 and 8 weeks after implantation using histologic and immunofluorescent analyses. The microscopic structure of the defected TA muscles with each group was observed by hematoxylin and eosin (H&E) and Masson's trichrome staining (MTS) (Fig. 5a). In the non-treated group, the surgically excised regions of the TA showed no sign of muscle regeneration, but fibrotic tissue was formed in the defect region, resulting in muscular atrophy. In the MPC only and MPC + NSC groups, the muscle tissue formation with minimum fibrotic tissue ingrowth was observed in the defect region, resulting in muscle volume maintenance. Quantitatively, the collagen deposition area (%) was measured with each group (Fig. 5b). More importantly, organized skeletal muscle development with aligned myofiber formation was observed in both the MPC only and MPC + NSC groups.

To investigate the contribution from the implanted hMPCs in the bioprinted skeletal muscle construct on muscle regeneration, the retrieved samples were evaluated using double-immunofluorescence for MHC and human leukocyte antigen (HLA) (Fig. 6). In both the MPC only and MPC + NSC groups, cells expressing HLA were visualized in the defected region at 4 and 8 weeks after implantation (Fig. 6a), while cells in the sham and non-treated groups did not express HLA. The result indicated that the implanted cells (HLA+) in the bioprinted constructs were able to differentiate and form the myofibers (MHC+/HLA+, white arrows), which contributed to muscle tissue regeneration with highly aligned and organized architecture. Moreover, the introduction of hNSCs in the bioprinted constructs (MPC + NSC) had a greater ability to form MHC+ myofibers when compared with the bioprinted constructs with hMPCs (MPC only), as confirmed by the % area of MHC+ myofibers (Fig. 6b) and % area of HLA+ myofibers (Fig. 6c). The

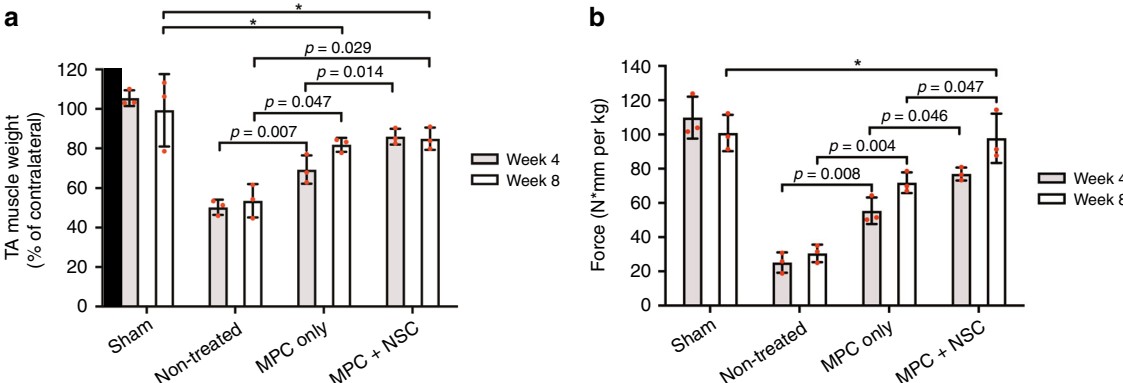

**Fig. 3 In vitro evaluation of 3D bioprinted human skeletal muscle constructs. a** Printing path for cell-laden hydrogel, PCL, and sacrificial material. **b** Bioprinted constructs containing hMPCs and hNSCs ($30 \times 10^6$ cells per ml, $10 \times 10 \times 3$ mm³ in dimension) after printing. **c** Live/dead staining images of the central part of bioprinted constructs containing MPC only and MPC + NSC (300:1) at 1 and 7 days. **d** Quantitative cell viability (%) at 1 day after printing ($n = 5$ per group). **e** Immunofluorescence for MHC (red) of MPC only and MPC + NSC. Quantitative analysis of **f** MHC⁺ myofiber density (%) ($n = 3$ per group) and **g** average length of MHC⁺ myofibers (μm) at 7 days ($n = 3$ per group, two-sided Student's $t$-test, *$p = 0.012$). **h, i** Immunofluorescence for (H) βIIIT (red) of differentiated neurons and (I) GFAP (red) of glial cells. The experimental findings were qualitatively reproduced three times. **j, k** Immunofluorescence for **j** AChR (green)/MHC (red) of AChRs pre-patterning on myotubes (white arrows) and **k** AChR (green)/βIIIT (red)/MHC (purple) of NMJs (white arrows). **l** Immunofluorescence for AChR (green)/MHC (red) of MPC only construct. **m** Number of AChRs per field ($n = 3$ per group). **n** Number of βIIIT⁺ AChR⁺ NMJs per field ($n = 3$ per group). All data are represented as mean ± SD. The $p$-values by two-sided Student $t$-test are indicated.

**Fig. 4 Functional restoration by 3D bioprinted human skeletal muscle constructs implanted in the rat model of tibialis anterior (TA) muscle defect at 4 and 8 weeks after implantation. a** TA muscle weight (% of contralateral normal TA muscle) ($n = 3$ per group and time point, triple measures per sample). **b** Tetanic force (N·mm per kg) ($n = 3$ per group and time point, triple measures per sample). All data are represented as mean ± SD. The $p$-values by two-way ANOVA followed by Tukey's test are indicated. *no significance.

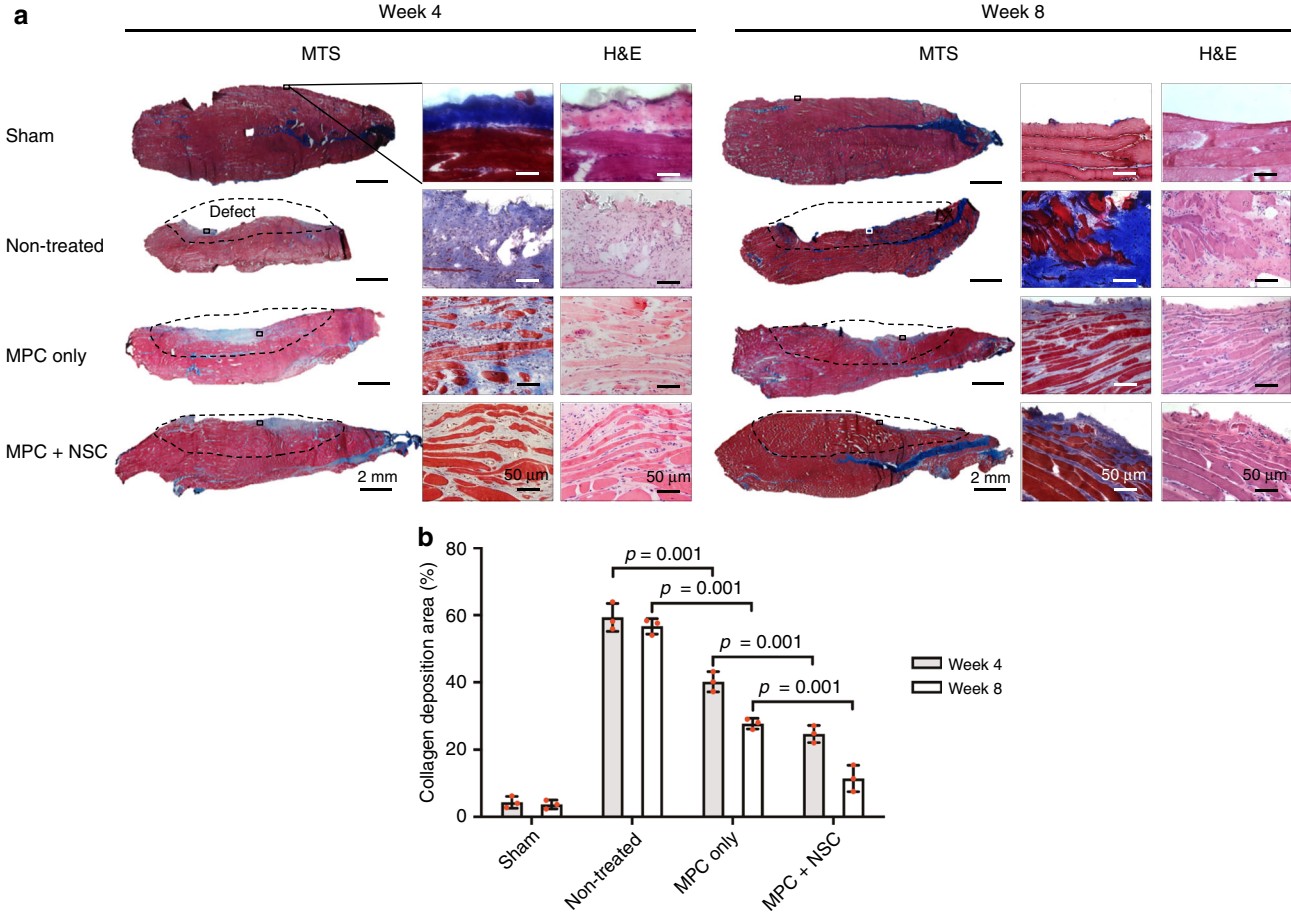

**Fig. 5 Histological examination. a** Skeletal muscle regeneration by 3D bioprinted constructs at 4 and 8 weeks after implantation. The MPC + NSC group showed aligned, newly formed muscle fibers with reduced fibrosis and maintenance of muscle volume. Dashed lines; defected area. MTS: Masson's trichome staining. H&E: hematoxylin and eosin. **b** Quantification of the area of collagen deposition (%) using MTS staining results ($n = 3$ per group and time point, four random fields per sample). All data are represented as mean ± SD. The $p$-values by two-way ANOVA followed by Tukey's test are indicated.

results indicated that the hMPCs in the bioprinted constructs could be matured and, eventually, formed thick myofibers while maintaining the tissue organization, and the neural cell component could enhance the skeletal muscle regeneration.

**NMJ formation**. To investigate the innervation capacity of the bioprinted skeletal muscle constructs, triple-immunofluorescence was performed for NF/AChR/MHC and NF/AChR/HLA. NMJs formation capacity of the implanted hMPCs in the 3D bioprinted muscle constructs was examined (Fig. 7a, b). In the MPC only group, co-localization of $NF^+$ axons of peripheral nerves and $AChR^+/MHC^+$ myofibers was observed in the implanted area, indicating NMJ formation between the host nerve and newly formed myofibers in the bioprinted muscle constructs. Moreover, the presence of $HLA^+$ cells along with $NF^+/AChR^+/MHC^+$ myofibers in the implanted region (yellow arrows) proved that the hMPCs were able to differentiate into myofibers and form NMJs that integrated with the host nerve. Notably, the MPC + NPC group showed mature NMJs and neuronal contact on the newly formed myofibers in the implanted constructs (Fig. 7a, b). Typical pretzel-like NMJs were observed in the MPC + NSC group that had similar morphology of NMJs in the sham group, whereas the MPC only group had inadequately developed NMJs at 8 weeks after implantation. In the MPC + NSC group, neuronal differentiation by the implanted hNSCs (white arrows,

$\beta IIIT^+/HLA^+$ neurons) and contact with host nerve ($\beta IIIT^+/HLA^-$ neurons) was also observed (Fig. 7c). This indicates the neuronal differentiation of implanted hNSCs and neural integration between host tissues and implanted constructs (yellow arrow; $\beta IIIT^+/HLA^-$ host neuron). Quantitatively, more NMJs per field and AChR clusters per field were found in the MPC + NSC group than in the MPC only group (Fig. 7d, e). Moreover, there was no significant difference in $NMJ^+$ myofibers (%) when compared to the sham group (Fig. 7f). These results indicated that interaction between muscle cells and neural cells in the bioprinted constructs improved NMJ formation ability on the bioengineered myofibers in vivo.

**Vascularization**. Vascularization of the implanted constructs was examined by double-immunofluorescent staining for Willebrand factor (vWF)/alpha smooth muscle actin (α-SMA). In the MPCs and MPC + NSC groups, the bioprinted muscle constructs were highly vascularized following implantation into the muscle defects, based on overall staining of $vWF^+/\alpha-SMA^+$ blood vessels in implanted constructs (Fig. 8a). The MPC only and MPC + NSC groups showed higher number and area of vessels compared with the non-treated group; however, there was no significant difference between the MPC only and MPC + NSC groups (Fig. 8b, c).

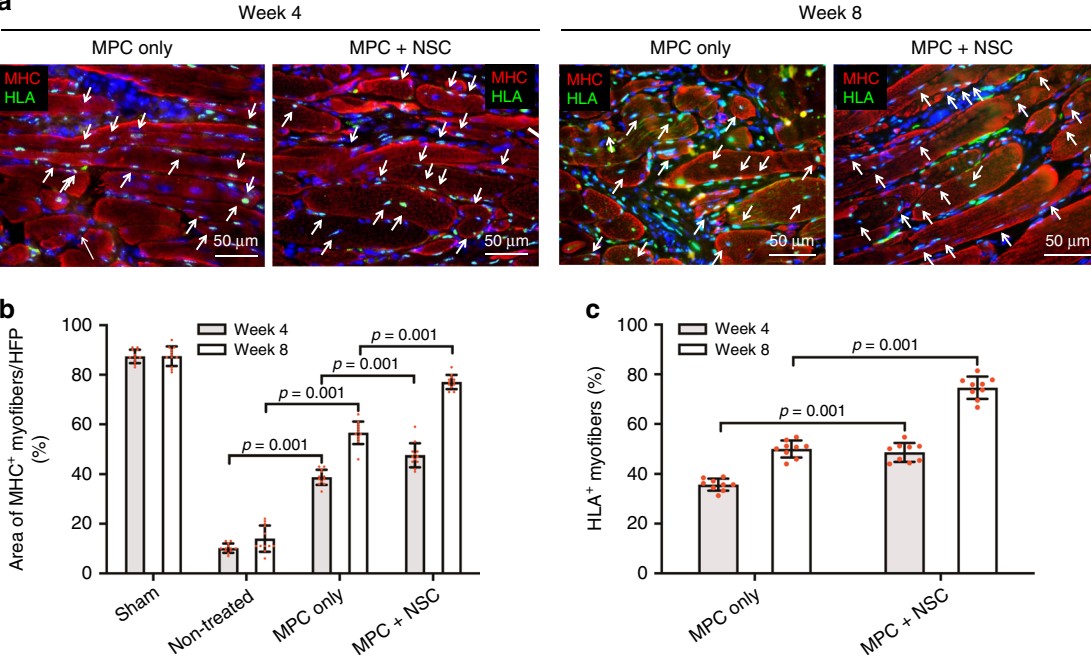

**Fig. 6 Myofiber formation and maturation. a** Immunofluorescence for MHC (red)/human leukocyte antigen (HLA, green) of injured TA muscles at 4 and 8 weeks after implantation. The presence of MHC$^+$ HLA$^+$ cells in the implanted region indicates that implanted human MPCs survived and contributed to newly formed muscle fibers (white arrow). **b** Quantification of the area of MHC$^+$ myofibers per field (%, ×400) ($n = 3$ per group and time point, four random fields per sample). **c** Quantification of HLA$^+$ myofibers (%, ×400) ($n = 3$ per group and time point, three random fields per sample). All data are represented as mean ± SD. The $p$-values by two-way ANOVA followed by Tukey's test are indicated.

## Discussion

Since skeletal muscle defects of >20% of mass lose the regenerative capacity[1], therapeutic options for these extensive muscle defect injuries are limited. As such, a bioengineered tissue construct with structural and functional characteristics of native skeletal muscle would lead to repairing muscle deformity, resulting in the restoration of muscle function. Recent advances in musculoskeletal tissue engineering enabled to generate cellularized tissue constructs that recapitulate the organization of native skeletal muscle containing aligned multinucleated muscle cells, leading to contractile function[2]. In this study, we fabricated the NMJ-preformed human skeletal muscle constructs by the addition of the neural cell component in the bioprinted constructs. The results show that the addition of hNSCs to the 3D bioprinted skeletal muscle constructs strongly suggests the accelerated differentiation and maturation of hMPCs, long-term survival, and effective NMJ formation with AChRs clustering in vitro. Furthermore, implantation of these bioprinted skeletal muscle constructs was able to facilitate the accelerated innervation and restore normal muscle anatomy and function in the rat TA muscle defect.

For the success of the bioengineered skeletal muscle constructs to restore the function of the injured muscle in vivo, rapid innervation with the host nerve is critical. Since establishing innervation may take up to 12 weeks in vivo, developing a strategy to accelerate the innervation of the implanted constructs is significant[33]. Unfortunately, there is no strategy for enhancing or accelerating innervation of the bioengineered skeletal muscle constructs that have not been fully developed, nor has the innervation potential of the constructs been demonstrated. Although several 3D co-culture systems composed of skeletal muscle cells and motor neurons have been reported, they were primarily aimed for in vitro models of NMJ-related diseases[37–39,43]. In this study, we demonstrated that neural input into the 3D bioprinted skeletal muscle constructs

improved the differentiation and survival of the bioprinted cells, as well as enhanced innervation capacity by interactions between neural cells and skeletal muscle cells.

It has been reported that neurotrophic factors and neurotransmitters released from neural cells significantly contribute the skeletal muscle development, as well as NMJ formation, stability, and maturation[29–32,44,45]. Thus, several studies demonstrated that 2D neuron-muscle co-culture improved skeletal muscle maturation and NMJ formation in vitro[36,37,39]. A few studies have also shown that the presence of motor neurons and NMJ formation in the bioengineered skeletal muscle constructs improves differentiation and maturation of skeletal muscle cells and their contractile function in vitro[37–39]. These pre-formed AChR clusters as NMJs on muscle fibers can enhance direct contacts with host nerve system (innervation) in vivo[29]. It has been also showed that glial cells can support the NMJ formation and tissue maturation, as well as efficient action potential[46–48]. Conversely, a study states that co-culture with non-myogenic cells can reduce the levels of myogenic differentiation due to the interruption of muscle cell fusion by other cell types.

Therefore, we investigated the effect of the neural cells on skeletal muscle differentiation and maturation and the optimal ratio of hMPCs and hNSCs in the 2D co-culture system. The results showed that the optimal cell ratio (hMPCs:hNSCs = 300:1) improved skeletal muscle cell differentiation and long-term survival and facilitated neuronal differentiation and NMJ formation on the muscle fibers. However, the decrease of myofiber formation was observed when hNSC ratios were higher than 300:1. According to these results, we fabricated 3D bioprinted skeletal muscle constructs with the optimal cell ratio, and these constructs showed high cell viability and accelerated myogenic differentiation. The hNSCs in the bioprinted constructs were also differentiated into neurons and glial cells, and these neural cell components seemed to readily induce the NMJ formation on the bioengineered skeletal muscle tissue.

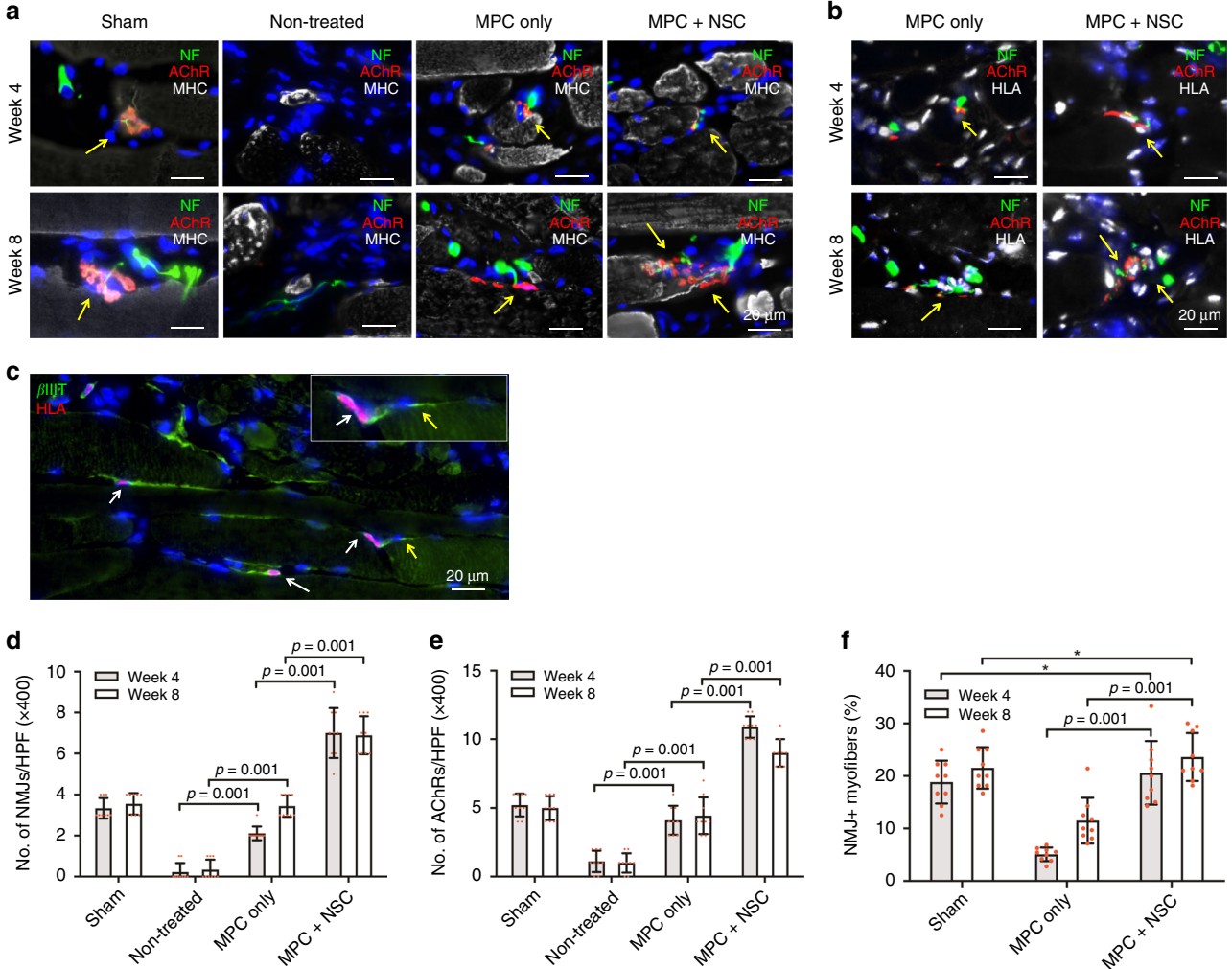

**Fig. 7 Neural integration with 3D bioprinted skeletal muscle constructs. a, b** Immunofluorescence for **a** NF (green)/AChR (red)/MHC (white) and **b** NF (green)/AChR (red)/HLA (white) of injured TA muscles at 4 and 8 weeks after implantation. **c** Immunofluorescence for βIIIT (green)/HLA (red) of the MPC + NSC constructs at 8 weeks after implantation. **d–f** Quantification of **d** number of NMJs per HPF (n = 3 per group and time point, three random fields per sample) and **e** number of AChRs per HPF (n = 3 per group and time point, three random fields per sample). **f** NMJ+ myofibers (%) (n = 3 per group and time point, three random fields per sample, *no significance). HPF high power field at ×400. All data are represented as mean ± SD. The p-values by two-way ANOVA followed by Tukey's test are indicated.

We hypothesized that neurotrophic factors released from neural cell components could contribute to survival, proliferation, and maturation of the bioengineered muscle tissues. We found that several factors presented at a high concentration in the hNSC-conditioned medium compared with the hMPC-conditioned medium (Supplementary Table 1). Such factors are known to improve myogenesis through promoting or enhancing muscle cell adhesion [e.g. vascular cell adhesion protein 1 (VCAP-1)[49]], proliferation [e.g. fibroblast growth factors (FGFs)[50,51], platelet-derived growth factors (PDGFs)[52,53], and osteopontin[54]], and differentiation and maturation [e.g. insulin-like growth factor binding protein-2 (IGFBP-2)[55] and insulin[56]]. Indeed, IGFBP-2 enhances muscle differentiation through facilitating growth arrest, a process required for the initiation of myoblast fusion, by enhanced Akt phosphorylation[55,57–59]. Insulin induces myogenesis through phosphatidylinositol (PI) 3-kinase/p70S6-kinase and p38-mitogen-activated protein kinase (MAPK) pathways[56].

In order to validate the NMJ-preformed bioprinted skeletal muscle constructs in vivo, these constructs were implanted in the TA muscle defect in rats. The in vivo outcomes showed effective nerve integration and vascularization of the implanted bioprinted

skeletal muscle constructs, resulting in accelerated restoration of muscle function. The construct innervation was confirmed by increasing numbers of NMJs and their levels of maturation, which were comparable to the age-matched sham control (native TA muscle). We previously demonstrated that the pre-fabrication of AChR clusters on bioengineered muscle tissue accelerated NMJ formation with host nerve integration after implantation[29]. Thus, we expected that the pre-formed AChRs on the MPC + NSC constructs (Fig. 3m) had a higher number of NMJs and AChRs after implantation compared with others (Fig. 7d). However, the muscle weights in both MPC only and MPC + NSC groups were not significantly different, indicating that the muscle mass was contributed from skeletal muscle cells rather than neural cell components. Although this present study did not include the contractibility of the bioprinted skeletal muscle construct in vitro, we speculated that neural input facilitated post-synaptic maturation and improved contractile force-generating capacity of the constructs in vivo. In-depth validation and refinement of the molecular, biological, and physiological aspects of the inner-vation and skeletal muscle regeneration process in the bioprinted constructs are currently being performed. Future studies will also

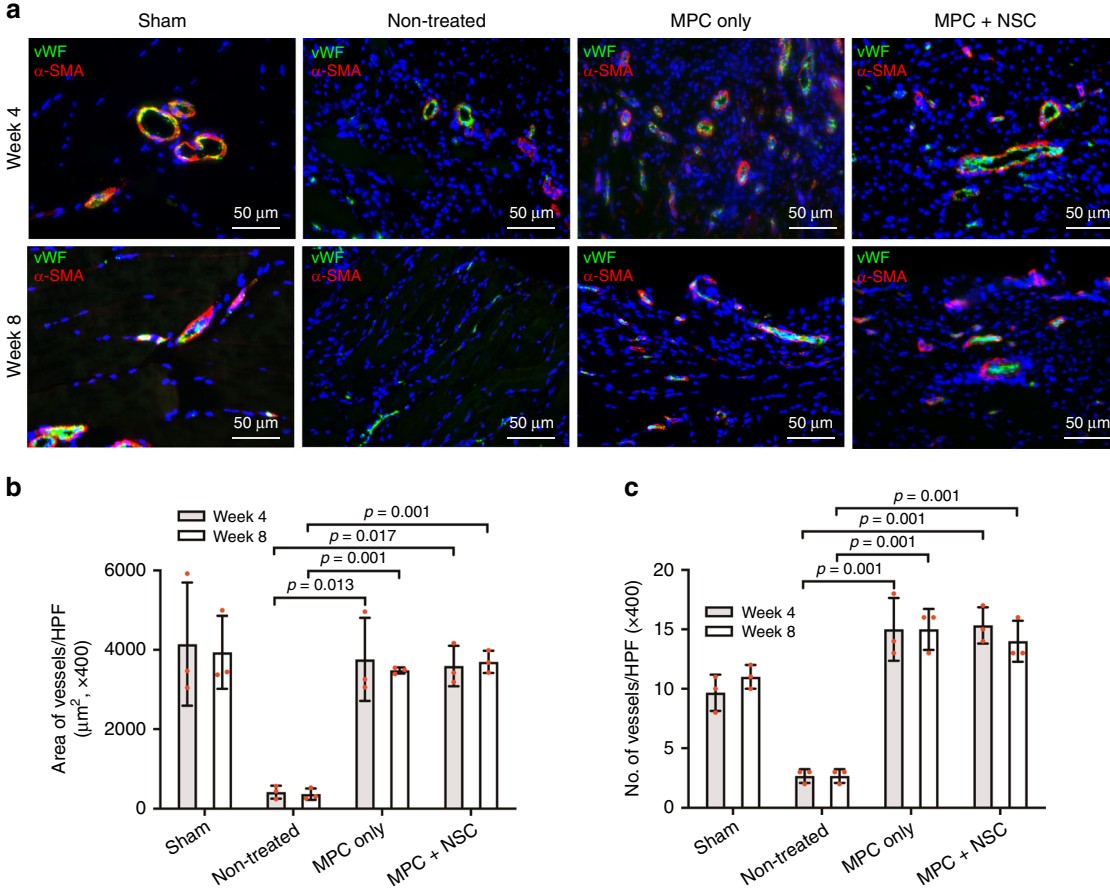

**Fig. 8 Vascularization of 3D bioprinted skeletal muscle constructs. a** Immunofluorescence for von Willebrand factor (vWF, green)/alpha smooth muscle actin (α-SMA, red) of injured TA muscles at 4 and 8 weeks after implantation. **b**, **c** Quantification of **b** area of vessels per HPF (μm²) (×400, $n = 3$ per group and time point, three random fields per sample) and **c** number of vessels per HFP (×400, $n = 3$ per group and time point, three random fields per sample). Bioprinted constructs implanted at the site of the muscle defect injury were vascularized, as confirmed by vWF$^+$/α-SMA$^+$ vessels, but there was no significant difference between the MPC only and MPC + NSC groups. All data are represented as mean ± SD. The $p$-values by two-way ANOVA followed by Tukey's test are indicated.

investigate the contractile function, such as twitch and tetanus of the bioengineered constructs.

Although this present study is promising to develop a new therapeutic option for reconstructive surgery, several challenges need to be overcome. For future translation, we have established an hMPC manufacturing system for autologous cell therapy applications. The safety and efficacy of using autologous hMPCs for cell-based therapies have been demonstrated in several clinical trials, including muscle cell therapy for the treatment of urinary incontinence (ClinicalTrials.gov Identifier: NCT01953315). The hMPC manufacturing platform has been expanded and integrated into the manufacturing system for these tissue-engineered skeletal muscle tissue products. However, this approach of harvesting, isolating, and expanding hNSCs may not be clinically practical. Alternate approaches could include using induced pluripotent stem cell (iPSC)-derived neural cells or biological factors that replace the functions of hNSCs to achieve similar cellular effects that lead to successful innervation. Vascularization is another critical aspect of bioengineered skeletal muscle constructs. Promoting vascularization of large volume tissue constructs for treating extensive muscle defect injuries is even more important to overcome the diffusion limits of oxygen and nutrients, and maintain the cell survival and function[37,44,45]. To overcome this limitation, we have been utilizing this 3D bioprinting strategy to bioengineer vascularized skeletal muscle constructs for pre-clinical and clinical applications. Lastly, this

study utilized immunocompromised rats for the construct implantation containing human-derived cells to determine the feasibility of bioengineered human skeletal muscle constructs for restoring skeletal muscle function in vivo. Although tissue engineering strategy utilizes autologous cell source derived from patients' own cells to avoid host immune response[60,61], future studies will need to investigate host responses, including inflammatory and immune responses and foreign body reaction, in the regenerative process using autologous cells in immunocompetent animals.

In conclusion, the neural cell component can support the long-term cell survival, enhance myogenic differentiation, and induce NMJ formation on the muscle fibers in the bioprinted skeletal muscle constructs in vitro, resulting in rapid restoration of muscle function in the rat TA muscle defect model. With further advances, this NMJ preformed bioengineered skeletal muscle construct may be an effective therapeutic approach for repairing extensive skeletal muscle defect injuries with accelerated innervation capacity.

## Methods

**Cell cultures.** Human MPCs were collected from biopsies of human gracilis muscles (from 51- and 64-year-old women, de-identified)[46]. All the procedures were approved by the Wake Forest University Institutional Review Board (IRB)-approved protocol. Donor tissue biopsies (average 200 mg samples) were rinsed in phosphate-buffered saline (PBS), and fat and other tissues were completely removed. The tissues were minced to less than $1 \times 1$ mm² and then digested in a

solution of 0.2% collagenase type I (Worthington Biochemical, Lakewood, NJ) and 0.4% dispase (Gibco, Grand Island, NY) in DMEM for 2 h at 37 °C. The digested tissues were washed with medium consisted of Dulbecco's Modified Eagle Medium (DMEM)/F12 nutrient Mix (1:1) (Gibco), 18% fetal bovine serum (FBS, Vally Biomedical Inc., Winchester, VA), 10 ng per ml human epidermal growth factor (EGF) (Millipore Sigma, Saint Louis, MO), 1 ng per ml human basic fibroblast growth factor (bFGF) (Millipore Sigma), 10 μg per ml human insulin (Millipore Sigma), and 0.4 μg per ml dexamethasone (Millipore Sigma). The obtained muscle fibers were gently pipetted, filtered through a strainer with a pore size of 100 μm, and centrifuged at 1500 rpm for 5 min. The pellets were resuspended in the medium and then transferred into 35-mm culture dishes coated with collagen type I (1 mg per ml, BD, Clontech, Bedford, MA). After incubation overnight in humidified atmospheric air (5% $CO_2$) at 37 °C, non-adherent cells were collected and transferred into collagen type I-coated 35-mm culture dishes. After transferring, the medium was changed at 4 and 7 days. Once the cells reached 80% confluence at 8–10 days, they were subcultured in a 10-cm culture dishes (passage 1). The isolated hMPCs were expanded in a growth medium composed of DMEM/high glucose (Thermo Scientific Inc., Waltham, MA), 20% FBS, 2% chicken embryo extract (Gemini Bio-Products, West Sacramento, CA), and 1% penicillin/streptomycin (P/S, Thermo Scientific). The hMPCs were expanded up to passage 4 for the experiments. Human NSCs (ReNcell VM) were obtained from EMD Millipore (Temecula, CA) and expanded to passage 4–6 in a ReNcell NSC maintenance medium consisted of 20 ng per ml EGF and 20 ng per ml bFGF, following the manufacturer's instructions.

For co-culture, cell suspensions containing both hMPCs and hNSCs with ratios from 1:0 to 500:1 were plated on laminin-coated 8-well glass chambers with the hMPC density of $6 \times 10^4$ per well, incubated overnight, and differentiated in co-culture medium for up to 10 days. The number of hMPCs was the same with different ratios with hNSCs. The differentiation medium consisted of DMEM/high glucose, 2% horse serum (Gibco), 1% ITS (Lonza, Basel, Switzerland), 250 nM dexamethasone (Millipore Sigma), and 1% P/S. As the hNSCs differentiation medium, the ReNcell maintenance medium without growth factors was used. The medium was changed every 3 days. For the human growth factor array, hMPCs and hNSCs ($1.5 \times 10^6$ per 15-cm culture dish) were plated in the growth medium, respectively. After 1 day, the plates were replaced with 20 ml of DMEM containing 1% P/S and cultured for 9 days. The conditioned media were collected and filtered through a 0.22 μm filter, and then concentrated (4×) using centrifugal filter units with a molecular weight cutoff (MWCO) of 3 kDa (Amicon Ultra-15 Centrifugal filter units, Merck Millipore, Darmstadt, Germany). The concentrated conditioned media were analyzed for growth factors and cytokines content using the Human Cytokine Array kit (RayBiotech, Norcross, GA)[62].

**Bioink preparation**. Three types of bioinks were applied to bioprint the skeletal muscle constructs: cell-laden bioink, sacrificing acellular bioink, and supporting polycaprolactone (PCL) bioink. To print cells, a fibrinogen-based composite hydrogel was prepared with 20 mg per ml fibrinogen (Millipore Sigma), 35 mg per ml gelatin (Millipore Sigma), 3 mg per ml hyaluronic acid (HA, Millipore Sigma), and 10% glycerol (Millipore Sigma) in DMEM/high glucose[2,25]. Briefly, HA was dissolved in DMEM/high glucose at 37 °C with stirring overnight, and then glycerol was added to the HA solution and stirred for 1 h. Fibrinogen and gelatin were added to the solution and dissolved at 37 °C for 1 h. After filtration through 0.45 μm syringe filters (Thermo Scientific), the cells were mixed through gentle pipetting. The sacrificing acellular bioinks were prepared by dissolving 35 mg per ml gelatin in DMEM/high glucose containing 3 mg per ml HA and 10% glycerol, followed by filtration. As a supporting PCL bioink, thermoplastic PCL polymer (Mw; 43,000–50,000, Polysciences, Inc., Warrington, PA) was used for the polymeric pillar structure.

**Three-dimensional bioprinting of skeletal muscle constructs**. The skeletal muscle constructs were fabricated using our custom bioprinting system[2,25]. The system consists of four dispensing modules (including cooling and heating units), a pneumatic pressure controller, XYZ stage and controller, a temperature controller, and a humidifier in a closed chamber. For fabrication of 3D skeletal muscle constructs, cell-laden hydrogel, acellular sacrificing hydrogel, and supporting polymer were used (Figs. 2 and 3a). Cell-laden and sacrificing acellular bioinks were loaded into different sterile syringes, which were connected to 300-μm Teflon® nozzles. The syringes were aseptically inserted into dispensing modules, which were connected to the pneumatic pressure controller. The PCL polymer was loaded into a stainless-steel syringe connected to a 300-μm metal nozzle and heated up to 95 °C during the printing process. The cell-laden bioink was printed at a speed of 90 mm per min through 50–70 kPa of pressure. The gelatin-sacrificing bioink was dispensed at 160 mm per min and 50–80 kPa. Feed rate and pressure of the PCL bioink was 75 mm per min and 780 kPa, respectively. Cell-laden bioink was printed in a parallel pattern in a layer-by-layer fashion. The PCL pillar structure was also printed simultaneously in each layer that anchors the printed cell-laden muscle bundles. Sacrificing gelatin-based bioink were printed between the printed muscle bundles, and then dissolved out at 37 °C after printing to create empty microchannels. The temperature of the closed aseptic chamber was maintained at 18 °C during the printing process. Printed constructs were cross-linked by thrombin solution (20 UI per ml, Millipore Sigma) for 30–60 min. The constructs were

cultured in the growth medium overnight and then switched in the differentiation medium supplemented with aprotinin (20 μg per ml, Millipore Sigma). The medium was changed every 3 days. For in vitro evaluations, the printed constructs having hMPCs only (1:0) and hMPCs and hNSCs (MPC + NSC, 300:1) were bioprinted ($10 \times 10^6$ per ml of cell density and $10 \times 7 \times 3$ mm$^3$ in dimension). For in vivo implantation, the bioprinted skeletal muscle constructs with a cell density of $30 \times 10^6$ per ml were prepared and cultured in vitro for 4–5 days in the differentiation medium before implantation[2].

**In vitro biological evaluation**. In vitro cellular activities and myogenic differentiation of hMPCs and hNSCs in the 2D co-culture and the 3D bioprinted constructs were examined. For immunofluorescent staining, all 2D and 3D samples were fixed with 4% of paraformaldehyde for 15–30 min and permeabilized in methanol at −20 °C for 10 min. The samples were blocked using a serum-free blocking agent (Dako, Carpentaria, CA) at room temperature for 1 h and incubated with primary antibodies for 1 h and the secondary antibodies for 40–60 min. To evaluate the myogenic differentiation of the hMPCs, the samples were stained with mouse anti-MF-20 antibody (1 μg per ml, Developmental Studies Hybridoma Bank, Iowa City, IA), mouse anti-myoD (1:200 dilution, Thermo Scientific) and rabbit anti-myogenin (1:200 dilution, Abcam, Cambridge, MA). Neuronal and glial differentiation of hNSCs were also evaluated by immunofluorescence for rabbit anti-βIIIT (1:100 dilution, Abcam), rabbit anti-NF (1:80 dilution, Millipore Sigma), and rabbit anti-GFAP (1:100 dilution, Abcam), respectively, following permeabilization in 0.1% Triton X-100 for 20 min. AChR clustering on the myotube was visualized for rat anti-AChR antibody (1:100 dilution, Abcam). NMJ formation was confirmed by double-immunofluorescence for anti-MF-20/AChR/βIIIT antibodies or anti-MF-20/AChR/NF antibodies. For the secondary antibodies, Texas Red-conjugated anti-mouse, anti-rabbit, or anti-rat IgG (1:200 dilution, Vector Labs, Burlingame, CA), Alexa 488-conjugated anti-rabbit or anti-rat IgG (1:200 dilution, Invitrogen, Eugene, OR), or Cy5-conjugated anti-mouse IgG (1:200 dilution, Invitrogen) were used. The samples were mounted with VECTASHEILD Mounting Medium with DAPI (Vector Labs), or Prolong® Gold Antifade Mountant (Life Technologies, Carlsbad, CA) followed by treated with DAPI (1:1000 dilution, Life Technologies) for 10 min. All antibodies were diluted with antibody diluent (Dako). Stained tissues were analyzed with confocal microscopes (FV10i, FV10-ASW 04.02., Olympus, Tokyo, Japan; Leica TCS LSI Macro Confocal, LAS-AF 3.1.8976.3, Leica, Microsystems, Wetzlar, Germany) or a fluorescent microscope (DM4000, Image Pro 6.3., cellSens Dimension 1.18, Leica). In the 2D culture, MHC$^+$ myotube density of each group was measured by using immunofluorescent images for MF-20 (×50 magnification), and the value was normalized by that of the MPC only group in a blinded fashion ($n = 3$ per sample). Myotube formation in the bioprinted constructs was evaluated as MHC$^+$ myotube density (%) and the average length of MHC$^+$ myotubes (μm) (×100 magnification, $n = 3$ per sample). All images were analyzed with Image J software (National Institutes of Health, Bethesda, MD).

The cell viability was determined using a live/dead assay/cytotoxicity kit (Life Technologies) according to the manufacturer's instructions. Briefly, the bioprinted constructs were incubated in the assay solution (50 μl per ml Cal-AM and 2 μl per ml Et-D in DMEM/high glucose) at room temperature for 40–60 min. After gentle washing with PBS, live and dead cells were imaged using a confocal microscope (Leica TCS LSI Macro Confocal). The number of live cells (green) and dead cells (red) were manually counted, and cell viability (%) was calculated ($n = 5$). To examine the functionality of NMJs on bioprinted muscle constructs, calcium uptake imaging was performed. After 5 days of differentiation, the constructs were treated with a calcium indicator (Fluo-4 Calcium Imaging Kit, Invitrogen) following the manufacturer's instruction. Fluorescent images were obtained after treatment with 500 mM acetylcholine (ACh, Millipore Sigma)[29,63–65].

**The rat TA muscle defect model**. The skeletal muscle defect injuries were created in RNU rats (male, 10–12-week-old, Charles River Laboratory, Wilmington, MA)[2,41,42] in accordance with a protocol approved by the Institutional Animal Care and Use Committee (IACUC) at Wake Forest University. Rats were anesthetized with 3% isoflurane, received a long incision on the skin of the left lower leg, and the muscles were separated from the fascia. The EDL and EHL muscles were removed to exclude compensatory hypertrophy during muscle regeneration, and approximately 40% of the middle third of the TA muscle was excised and weighed. TA muscle weight of each animal was calculated by the following equation: $y(g) = 0.0017 \times$ body weight (g) $- 0.0716$[41]. After removing PCL pillars, the bioprinted constructs ($10 \times 7 \times 3.6$ mm$^3$ in dimension), which fitted the defect region in TA muscle) were implanted in the muscle defect sites and covered with fascia. Fascia and skin were closed using sutures and surgical staples, respectively. In this study, 4 groups were studied at 4 and 8 weeks after implantation (total 24 rats, $n = 3$ per each group and each time point): (1) sham (without defect), (2) non-treated (defect only), (3) MPC only (bioprinted skeletal muscle construct containing hMPCs), and (4) MPC + NSC [bioprinted muscle construct containing hMPCs and hNSCs (300:1)].

**In vivo functional examination**. Muscle function of the injured TA muscle was evaluated at 4 and 8 weeks after implantation by measuring tetanic force of TA

muscle with a dual-mode muscle lever system (Aurora Scientific, Inc., Mod, 305b, MI-RAT 2.74, Aurora, Canada) in blinded fashion ($n = 3$ per group and time point, three repeated measurements per sample)[2,42]. Under anesthesia, the left foot of each rat was attached to a footplate, and the knee and ankle were stabilized at a 90° angle. Two sterilized platinum needle electrodes were placed in the posterior compartment of the lower leg along either side of the peroneal nerve, and the nerve was stimulated using a Grass stimulator (S88) at 100 Hz and 10 V with a pulse width of 0.1 ms. Muscle force (N·mm per Kg) in response to the electrical stimulation was calculated by (peak isometric torque × foot length) per body weight. After the functional assessment, TA muscles of each lower leg were isolated and weighed, and the percentage of TA muscle weight (% of contralateral) was calculated ($n = 3$ per group and time point).

**Histologic and immunofluorescent analyses**. For histologic evaluation of in vivo samples, harvested TA muscle tissues were freshly frozen in liquid nitrogen and cryosectioned into 6-μm thick slices. After fixation with 4% paraformaldehyde for 10 min, H&E and Masson's trichrome staining were performed. For immunofluorescence, the fixed tissue sections were treated in methanol at −20 °C for 15 min, in a serum-free blocking agent for 1 h at room temperature, and then incubated with primary antibodies for 1 h. The tissue sections were incubated in the secondary antibodies such as Texas Red-conjugated anti-mouse or anti-rat or IgG, Alexa 488-conjugated anti-rabbit or anti-chick (1:200 dilution, Invitrogen) IgG, or Cy5-conjugated anti-mouse or anti-rabbit (1:200 dilution, Invitrogen) IgG for 40–60 min, and mounted with Mounting Medium with DAPI.

To examine the newly formed muscle fibers in the implantation regions, tissue sections were double-immunostained with mouse anti-MF-20 and rabbit anti-HLA-A (1:100 dilution, Abcam). Area of MHC$^+$ myofibers per field (%) and HLA$^+$ myofibers (%) were measured with immunofluorescent images for MHC/HLA (×400 magnification) in a blinded fashion ($n = 3$ per sample and time point, 3–4 random fields in each sample). For evaluating vascularization of implanted constructs, tissue sections were stained with rabbit anti-vWF (1:400 dilution, Dako) and mouse anti-α-SMA (1:50 dilution, Santa Cruz Biotechnology, Santa Cruz, CA). The number and area of vessels per field (μm$^2$) were measured with immunofluorescent images for vWF/α-SMA (×400 magnification, $n = 3$ per sample and time point). Neurons and NMJ formation of the implanted constructs were visualized by immunofluorescence with rabbit anti-NF/rat anti-AChR/mouse anti-MF-20, chicken anti-NF (1:1000 dilution, Abcam)/rat anti-AChR/rabbit anti-HLA, and mouse anti-HNA (human nuclear antigen, 1:100 dilution, EMD Millipore)/rabbit anti-βIIIT antibodies. Using triple-immunofluorescence for NF/AChR/MHC (×400 magnification), numbers of NMJ per field and AChR per field and NMJ$^+$ myofibers (%) were counted ($n = 3$ per group and time point, four random fields per sample).

**Statistical analysis**. Results were analyzed with Origin Pro 8.5 (OriginLab Co, Northampton, MA), GraphPad Prism 8 (GraphPad Software, San Diego, CA), and SPSS software (SPSS, version 19; IBM, Armonk, NY). One-way or two-way analysis of variance (ANOVA), Tukey's post hoc tests, and Student's $t$-test were applied to mean comparisons. Variables are expressed as a mean ± standard deviation (SD), and differences between experimental groups were considered statistically significant at $p < 0.05$.

**Reporting summary**. Further information on research design is available in the Nature Research Reporting Summary linked to this article.

## Data availability
The data has been included in the manuscript or Supplementary Information. The source data underlying Figs. 1b–e, i, j, 3d, f, g, m, n, 4a, b, 5b, 6b, c, 7e, f, and 8b, c, Supplementary Figs. 1b, d, and 2c, d are provided as a Source Data file. The additional data generated or analyzed during this study are available from the corresponding author on reasonable request.

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

## Acknowledgements
This work was supported by the Army, Navy, NIH, Air Force, VA, and Health Affairs to support the AFIRM II effort under Award No. W81XWH-14-2-0004. The U.S. Army Medical Research Acquisition Activity, 820 Chandler Street, Fort Detrick MD 21702-5014 is the awarding and administering acquisition office. Opinions, interpretations, conclusions, and recommendations are those of the author and are not necessarily endorsed by the Department of Defense. This work was, in part, supported by the Medical Technology Enterprise Consortium (MTEC) (grant no. 2017-614-002) and the State of North Carolina. We thank Ms. Margaret M. Vanschaayk for cell culture, Dr. Young Koo Lee for animal surgery, Regenerative Medicine Clinical Center (RMCC) for hMPCs isolation, and Ms. Karen Klein at the Wake Forest Clinical and Translational Science Institute (UL1 TR001420; PI: McClain) and Dr. Colin Bishop for editorial assistance.

## Author contributions
J.H.K., S.J.L., A.A., and J.J.Y. developed the concept of the bioprinted skeletal muscle constructs and designed all experiments. J.H.K and Y.J.S. performed in vitro experiments, and J.H.K and I.K. performed in vivo experiments. J.H.K., Y.J.S., I.K.K., I.K., J.J.Y., A.A., and S.J.L. analyzed data and J.H.K. wrote the manuscript. S.J.L., J.J.Y., and A.A. edited the manuscript and J.J.Y., A.A., and S.J.L. provided the direction of the project.

## Competing interests
The authors declare no competing interests.

## Additional information

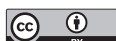

