## [Peer Review File · Nature Communications]

Reviewers' comments:

Reviewer #1 (Remarks to the Author):

The paper titled "Neural Cell Integration into 3D Bioprinted Skeletal Muscle Constructs Accelerates Restoration of Muscle Function" is a highly topical study. It reports outstanding quality results on inclusion of neuronal cells in bioprinting of skeletal muscle to rapidly integrate the bioprinted muscle in the host neuronal system and to accelerate the muscle regeneration. The authors report on the optimisation of the bioprinting of the muscle with an integrated neuronal component and implantation of the bioprinted muscle in an athymic nude (RNU) rat model for 8 weeks. The study reports some impressive results, including the accelerated formation of neuromuscular junctions to integrate the muscle with the host neuronal system, and evidence of rapid vascularization of the bioprinted nerve. The study also reports a functional recovery of the muscle by measuring muscle weight and muscle tetanic force measurement. The authors set their results well within the framework of recent literature and highlight a future direction of this research. The paper is expertly written and referenced and the figures are of high publication standard. The results of the study are of excellent quality and merit publication in Nature Communications. The general subject area of this study is highly topical and this study definitely sets a high benchmark for future work in this field. I can recommend publication of this study after the authors have considered a few small suggestions:

- 1) The authors could add a scale bar to Figure 3 A, or dimensions to Figure 2 B/C to give an indication of the size of the implant.
- 2) As I understand from the manuscript the PCL is used as a support material for the in vitro culture pre-implantation, but it was not part of the implant. Where exactly were the pillars positioned, were the pillars on the outside of the bioprinted construct or were they also inside, as prongs, to retain the bioprinted construct in place. A slightly more detailed description of the bioprinting process and the role of the PCL pillars would help.
- 3) Figure 5 shows clearly a striated structure for the bioengineered muscles, but there are also clearly ~10 μm gaps observable, which I assume are produced by the remnants of the sacrificial material to make the bioprinted muscle. There seems to be a distinct difference in density of the muscle tissue definitely in between week 4 and 8 MP+NSC and MPC only (Figure 5). Maybe the authors would like to comment on these observations.
- 4) The authors correctly state that "Host responses, including inflammatory response and foreign body reactions, in the regeneration process, need further investigation" but provide little detail. Do the authors expect any hostile host response to the implant, in particular if the suggested cellular components are progenitors (hMPCs) or stem cells (hNSCs). A very short discussion could be added.
- 5) Figure 6C x-axis label "myofiebrs" should be "myofibers".

Reviewer #2 (Remarks to the Author):

The authors present a bioengineered skeletal muscle construct embedding human muscle progenitor cells and human neural stem cells. Muscle progenitor cells differentiate into aligned myotubes while neural stem cells contribute to the generation of neuronal and glial populations. The presence of neural stem cells is shown to increase the in vitro maturation and long term survival of the bioprinted skeletal muscle constructs. Moreover, the integration of neural cells promoted early signs of innervation and restored the muscle weight and contractility in a rodent model of tibialis anterior defect.

The authors clearly emphasize in the introduction the importance of engineering muscle tissue constructs that could restore the functionality of irreversibly damaged muscles. Several recent advancements in the field are presented. However, the authors should also consider other milestones, including but not limited to:

- Levenberg, Rouwkema et al. Nature Biotech, 2005
- Shandalov, Egozi et al. PNAS, 2014

In addition, the authors do not mention important contributions in the field of bioengineered neuromuscular junctions:

- Uzel et al., Science Advances, 2016
- Dixon et al., Tissue Engineering C, 2018

And recently developed models of vascularized muscle with endomysium:

- Bersini et al., Cell Reports, 2018

The result section presents several characterizations of the bioprinted construct, both in vitro and in vivo. However, many aspects are not clear and they would need more data/explanations to go beyond a simple incremental study.

Here are some specific comments/questions (in order of appearance in the manuscript):

- In the discussion of the effect of neural cells on myotube formation, it is not clear if the same number of human muscle progenitor cells was used in all conditions where the cell ratio was tuned. Is the number of muscle progenitor cells the same comparing 1:0 and 300:1 conditions?
- The MHC staining is not always convincing. Additional staining of markers of muscle differentiation and maturation are strongly recommended
- The authors present several images of the bioengineered construct, however these images should always be coupled with quantifications (e.g. images presented in Fig. 1 (quantification of co-localizations), Fig.5 (quantification of MTS))
- In Figure 1, are the authors considering single slices or projected 3D stacks? Images are not exhaustive and electron microscopy images are strongly recommended to prove the presence of neuromuscular junctions
- In the quantification of Live&Dead assay, muscle+neural cell samples showed an absolute value of 94.99% viability (and not a 94.99% increase in cell viability compared to constructs embedding muscle cells only)
- Figure 3D: it is not clear why the fiber is much thicker in the co-culture condition
- Figure 3J: quantification is encouraged. Moreover, images are not clear and the color choice is questionable because it does not allow to correctly discriminate each marker. Also, which is the difference between co-culture and monoculture constructs? Quantification is required. Finally, electron microscopy images would provide a clearer picture of the differences between the two conditions
- Figure 4: statistical differences are not well explained (also true for Figure 6).
- Do bioengineered muscles contract in vitro? Is there any difference in the co-culture vs. monoculture conditions in response to electrical stimulation? This aspect would be really important when discussing the formation of neuromuscular junctions
- Figure 5: it is not clear if the enlarged images come from the same anatomical region within the defect created in the tibialis anterior. Also, it is not clear if the tissue sections are collected from muscle samples with the same orientation. Finally, the authors should clearly highlight with dashed lines the defect in the muscle samples
- The section "Host nerve integration of the 3D bioprinted skeletal muscle constructs" presents conclusions which are not fully supported by the data provided in Fig. 7

Some additional general questions are:

- How do the authors explain that co-culture constructs have more neuromuscular junctions and AChR clusters than the Sham group? Is it physiological?
- The bioengineered constructs are contributing to muscle regeneration, but are the integrating nerves functional? Is the presence of functional nerves correlating with the improved muscle function (i.e. force generation)?

- The authors discuss that factors secreted by differentiating neural stem cells might contribute to muscle maturation. Which factors? Which is the biological mechanism?
- An useful control for all the experiments would be an acellular construct with an empty ECM

The authors conclude the manuscript with a long discussion emphasizing that additional validations and refinements are required to completely characterize the bioengineered muscle constructs. Most of these refinements would be necessary to go beyond an incremental study for the field of muscle tissue engineering.

There are minor grammar errors and sentences which are not correctly formulated, including but not limited to:

"In this study, we investigated the feasibility using the bioprinted neural cell-integrated..."

"We evaluated the effect of neural cells in the bioprinted constructs with the ratio of hMPCs and NSCs for aspects of viability..."

RESPONSE TO REFEREES LETTER

Reviewer #1: The paper titled “Neural Cell Integration into 3D Bioprinted Skeletal Muscle Constructs Accelerates Restoration of Muscle Function” is a highly topical study. It reports outstanding quality results on inclusion of neuronal cells in bioprinting of skeletal muscle to rapidly integrate the bioprinted muscle in the host neuronal system and to accelerate the muscle regeneration. The authors report on the optimisation of the bioprinting of the muscle with an integrated neuronal component and implantation of the bioprinted muscle in an athymic nude (RNU) rat model for 8 weeks. The study reports some impressive results, including the accelerated formation of neuromuscular junctions to integrate the muscle with the host neuronal system, and evidence of rapid vascularization of the bioprinted nerve. The study also reports a functional recovery of the muscle by measuring muscle weight and muscle tetanic force measurement. The authors set their results well within the framework of recent literature and highlight a future direction of this research. The paper is expertly written and referenced and the figures are of high publication standard. The results of the study are of excellent quality and merit publication in Nature Communications. The general subject area of this study is highly topical and this study definitely sets a high benchmark for future work in this field. I can recommend publication of this study after the authors have considered a few small suggestions:

- 1) The authors could add a scale bar to Figure 3A, or dimensions to Figure 2B/C to give an indication of the size of the implant.

Response: We appreciate the reviewer’s comment. We have added a scale in Figure 3.

- 2) As I understand from the manuscript the PCL is used as a support material for the in vitro culture pre-implantation, but it was not part of the implant. Where exactly were the pillars positioned, were the pillars on the outside of the bioprinted construct or were they also inside, as prongs, to retain the bioprinted construct in place. A slightly more detailed description of the bioprinting process and the role of the PCL pillars would help.

Response: We appreciate the reviewer’s comment. The detailed information, including design concept, printing path, and bioprinting process, has been added in the Methods section. We have also added the printing path as Figure 3A for a better understanding. The PCL pillar only presented on the outside of the construct to support the cell-laden construct. For the implantation, the PCL structure was removed from the construct and only cellular structure was implanted.

- 3) Figure 5 shows clearly a striated structure for the bioengineered muscles, but there are also clearly $\sim 10\ \mu\text{m}$ gaps observable, which I assume are produced by the remnants of the sacrificial material to make the bioprinted muscle. There seems to be a distinct difference in density of the muscle tissue definitely in between week 4 and 8 MPC+NSC and MPC only (Figure 5). Maybe the authors would like to comment on these observations.

Response: We appreciate the reviewer’s comment. We assumed that the gaps between newly formed myofibers might be the undifferentiated muscle cells in the construct. In

Figure 6B, the area of myofibers/HFP increased over time. This could be evidence that the bioengineered muscle with NSCs accelerated its maturation and development.

- 4) The authors correctly state that “Host responses, including inflammatory response and foreign body reactions, in the regeneration process, need further investigation” but provide little detail. Do the authors expect any hostile host response to the implant, in particular if the suggested cellular components are progenitors (hMPCs) or stem cells (hNSCs). A very short discussion could be added.

Response: We appreciate the reviewer’s comment. According to the reviewer’s comment, a short discussion has been added to the Discussion.

- 5) Figure 6C x-axis label “myofiebrs” should be “myofibers”.

Response: The typo has been corrected. Thank you for the correction.

Reviewer #2: The authors present a bioengineered skeletal muscle construct embedding human muscle progenitor cells and human neural stem cells. Muscle progenitor cells differentiate into aligned myotubes while neural stem cells contribute to the generation of neuronal and glial populations. The presence of neural stem cells is shown to increase the in vitro maturation and long term survival of the bioprinted skeletal muscle constructs. Moreover, the integration of neural cells promoted early signs of innervation and restored the muscle weight and contractility in a rodent model of tibialis anterior defect.

1. The authors clearly emphasize in the introduction the importance of engineering muscle tissue constructs that could restore the functionality of irreversibly damaged muscles. Several recent advancements in the field are presented. However, the authors should also consider other milestones, including but not limited to:

- Levenberg, Rouwkema et al. Nature Biotech, 2005
- Shandalov, Egozi et al. PNAS, 2014

In addition, the authors do not mention important contributions in the field of bioengineered neuromuscular junctions:

- Uzel et al., Science Advances, 2016
- Dixon et al., Tissue Engineering C, 2018

And recently developed models of vascularized muscle with endomysium:

- Bersini et al., Cell Reports, 2018

Response: We appreciate the reviewer’s thorough literature review in the field of skeletal muscle tissue engineering. We have added the studies mentioned above to the Introduction.

The result section presents several characterizations of the bioprinted construct, both in vitro and in vivo. However, many aspects are not clear and they would need more data/explanations to go beyond a simple incremental study. Here are some specific comments/questions (in order of appearance in the manuscript):

2. In the discussion of the effect of neural cells on myotube formation, it is not clear if the same number of human muscle progenitor cells was used in all conditions where the cell ratio was tuned. Is the number of muscle progenitor cells the same comparing 1:0 and 300:1 conditions?

Response: We appreciate the reviewer's comment. The number of hMPCs was the same with different ratios with hNSCs. This has been added to the Methods section.

3. The MHC staining is not always convincing. Additional staining of markers of muscle differentiation and maturation are strongly recommended.

Response: We appreciate the reviewer's comment. We are quite sure that muscle differentiation can be confirmed by myotube formation with MHC expression. Without this morphological change, MHC expression did not occur. Per the reviewer's recommendation, additional immunofluorescence for myoD and myogenin was performed. The results have been presented in Supplementary Figure 3.

4. The authors present several images of the bioengineered construct, however these images should always be coupled with quantifications (e.g. images presented in Fig. 1 (quantification of co-localizations), Fig.5 (quantification of MTS)).

Response: We appreciate the reviewer's comment. Per the reviewer's comment, the quantification of the number of AChRs, the number of BIIT⁺ AChRs (co-localization), and collagen deposition (%) using MTS have been presented in Figure 1 and Figure 5.

5. In Figure 1, are the authors considering single slices or projected 3D stacks? Images are not exhaustive and electron microscopy images are strongly recommended to prove the presence of neuromuscular junctions.

Response: We appreciate the reviewer's thorough review and comment. Unfortunately, we had a technical difficulty to prepare 2D culture sample for the electron microscopic images. Alternatively, we have added 3D stacked confocal microscopic images to support the co-localization of β III tubulin⁺ neuron and AChR (NMJ formation) as shown in Supplementary Figure 4.

6. In the quantification of Live&Dead assay, muscle+neural cell samples showed an absolute value of 94.99% viability (and not a 94.99% increase in cell viability compared to constructs embedding muscle cells only).

Response: We appreciate the reviewer's comment. This has been corrected.

7. Figure 3D: it is not clear why the fiber is much thicker in the co-culture condition.

Response: We appreciate the reviewer's comment. The image of the MPC+NSC shows two printed cell-laden struts overlapped. In order to avoid any misinterpretation, the image has been modified.

8. Figure 3J: quantification is encouraged. Moreover, images are not clear and the color choice is questionable because it does not allow to correctly discriminate each marker. Also, which is the difference between co-culture and monoculture constructs? Quantification is required. Finally, electron microscopy images would provide a clearer picture of the differences between the two conditions.

Response: We appreciate the reviewer's comment. As the reviewer's recommendation, we have added the number of AChRs/HFP (Figure 3M). The quantification result showed that a higher number of AChRs were expressed in the co-culture than the monoculture constructs. In addition, we quantified the number of NMJs (β IIIIT⁺ AChR⁺)/HFP (Figure 3N). To clearly show the β IIIIT⁺ AChR⁺ NMJs, the construct was stained with only β IIIIT and AChRs using two colors (Green, β IIIIT; Red, AChR⁺) (Supplementary Figure 4).

9. Figure 4: statistical differences are not well explained (also true for Figure 6).

Response: We appreciate the reviewer's comment. We have improved the explanation of statistical differences in Figure 4 and Figure 6.

10. Do bioengineered muscles contract *in vitro*? Is there any difference in the co-culture vs. monoculture conditions in response to electrical stimulation? This aspect would be really important when discussing the formation of neuromuscular junctions.

Response: We appreciate the reviewer's comment. We agree that the muscle contractility *in vitro* in response to electrical stimulation can be an indicator of the functionality of NMJs. Even though we tried to measure the *in vitro* contractility of the bioprinted muscle constructs, the contractile force generated by the differentiated MPCs in the bioprinted muscle constructs was not sufficient to be measured. Since our approach aimed to develop the implantable muscle constructs to treat extensive muscle defect injuries, the bioprinted constructs were cultured *in vitro* only for 4-5 days in the differentiation medium before implantation. This could be a reason that the contractile force was not sufficiently generated by the bioprinted constructs. Alternatively, we performed calcium uptake imaging *in vitro* to validate the functionality of the NMJs in terms of synaptic transmission and calcium channels opening which results in muscle contraction. We observed the increased number of cells with a high level of intracellular calcium in the MPC+NSC constructs compared with the MPC only constructs (Supplementary Figure 5).

11. Figure 5: it is not clear if the enlarged images come from the same anatomical region within the defect created in the tibialis anterior. Also, it is not clear if the tissue sections are collected from muscle samples with the same orientation. Finally, the authors should clearly highlight with dashed lines the defect in the muscle samples.

Response: We appreciate the reviewer's comment. All the sections were collected from tibialis anterior muscle samples with the same orientation (longitudinal-section of the tibialis anterior muscle of each group). The original defect area of each group was highlighted with a dashed line, and the area where the enlarged images were obtained was indicated with a solid line.

12. The section "Host nerve integration of the 3D bioprinted skeletal muscle constructs" presents conclusions which are not fully supported by the data provided in Fig. 7.

Response: We agree with the reviewer's comment that our *in vivo* data may not provide sufficient information to support the host nerve integration of the bioprinted skeletal muscle constructs. However, we believe that the bioprinted muscle construct is able to form NMJs that accelerate host nerve integration (innervation) *in vivo*. We have replaced 'host nerve integration' with 'NMJ formation'.

13. How do the authors explain that co-culture constructs have more neuromuscular junctions and AChR clusters than the Sham group? Is it physiological?

Response: We appreciate the reviewer's comment. In our previous study, we demonstrated that the pre-fabrication of AChR clusters on bioengineered muscle tissue accelerated the innervation (NMJ formation) after implantation [Ko IK et al. *Biomaterials*. 2013;34:3213-3255]. In this present study, the addition of NSCs increased the number of AChRs in the MPC+NSC constructs *in vitro* before implantation as shown in Figure 3M. So, we expected that the pre-fabricated AChRs on the MPC+NSC constructs had a higher number of NMJs and AChRs compared with others, including the Sham control. This has been included in the Discussion.

14. The bioengineered constructs are contributing to muscle regeneration, but are the integrating nerves functional? Is the presence of functional nerves correlating with the improved muscle function (i.e. force generation)?

Response: We appreciate the reviewer's comment. For the *in vivo* functional examination (Figure 4B), the common peroneal nerve of the rat was electrically stimulated, and then the force of the dorsal reflection of the foot in response to the electrical stimulation was measured. The common peroneal nerve is innervated with the tibialis anterior muscle where the bioprinted construct is implanted, so we speculate that the increased muscle force in the bioprinted muscle construct compared with the non-treated (defect only) group is due to the functional integration between host nerves and the implanted construct. In addition, we think that the increased muscle force generation in the MPC+NSC compared with MPC only may result in the pre-formed neuromuscular junctions and their functional integrations with host nerves.

15. The authors discuss that factors secreted by differentiating neural stem cells might contribute to muscle maturation. Which factors? Which is the biological mechanism?

Response: We appreciate the reviewer's comment. Using human growth factors and cytokines array, we were able to detect several neurotrophic factors such as insulin-like growth factor binding protein-2 (IGFBP-2) and insulin that could improve muscle maturation and development (Supplementary Table 1). This has been included in the Discussion.

16. An useful control for all the experiments would be an acellular construct with an empty ECM.

Response: We appreciate the reviewer's comment. In our previous study, the bioprinted muscle construct (MPC only) was compared with acellular construct (gel only) and non-printed cellularized construct. The results showed that the MPC only group showed superior muscle function recovery and myofiber formation with organized architecture, while the other groups showed limited muscle function recovery and tissue development (Kim JH et al., Sci Rep. 2018;8:12307).

17. The authors conclude the manuscript with a long discussion emphasizing that additional validations and refinements are required to completely characterize the bioengineered muscle constructs. Most of these refinements would be necessary to go beyond an incremental study for the field of muscle tissue engineering.

Response: We appreciate the reviewer's comment. We have improved the description in the Discussion.

18. There are minor grammar errors and sentences which are not correctly formulated, including but not limited to:

"In this study, we investigated the feasibility using the bioprinted neural cell-integrated..."

"We evaluated the effect of neural cells in the bioprinted constructs with the ratio of hMPCs and NSCs for aspects of viability..."

Response: We appreciate the reviewer's comment. The grammatical errors have been corrected.

REVIEWERS' COMMENTS:

Reviewer #1 (Remarks to the Author):

I have checked the manuscript and the responses to the reviewers queries and they have been answered adequately. I recommend publication of the manuscript.

Reviewer #2 (Remarks to the Author):

The authors addressed most of the comments in the revised version of the manuscript. I still recommend to be more cautious regarding the section on "NMJ formation of the 3D printed skeletal muscle constructs" (previously "Innervation of the 3D printed skeletal muscle constructs"). Based on the material provided by the authors, I cannot see if there is a full integration between the host nerves and the implanted construct. Therefore, I recommend to be more clear both in the result and in the discussion sections.

REVIEWERS' COMMENTS:

Reviewer #1 (Remarks to the Author):

I have checked the manuscript and the responses to the reviewers' queries and they have been answered adequately. I recommend publication of the manuscript.

Response: We appreciated the reviewer's comment.

Reviewer #2 (Remarks to the Author):

The authors addressed most of the comments in the revised version of the manuscript. I still recommend to be more cautious regarding the section on "NMJ formation of the 3D printed skeletal muscle constructs" (previously "Innervation of the 3D printed skeletal muscle constructs"). Based on the material provided by the authors, I cannot see if there is a full integration between the host nerves and the implanted construct. Therefore, I recommend to be more clear both in the result and in the discussion sections.

Response: We appreciate the reviewer's comment on the host nerve integration of the implanted construct. As recommended, we have improved the section on "NMJ formation". Even though we observed the improved NMJ formation in the bioprinted MPC + NSC constructs compared with the MPC only constructs, this did not mean the full integration between the host nerves and the implanted construct. In this study, we confirmed the innervation of the bioengineered muscle constructs by co-localization of neurofilament and acetylcholine receptor on the myofibers and tetanic muscle force measurement that are most commonly used methods [1-6].

[References]

- [1] Gilbert-Honick J., et al., Vascularized and innervated skeletal muscle tissue engineering. *Adv Healthcare Mater* 9, 1900626, 2020
- [2] VanDusen K.W., et al., Engineered skeletal muscle units for repair of volumetric muscle loss in the tibialis anterior muscle of a rat. *Tissue Eng A* 20, 2920, 2014
- [3] Mintz E.L., et al., Long-term evaluation of functional outcomes following rat volumetric muscle loss injury and repair, DOI: 10.1089/ten.TEA.2019.0126
- [4] Wu X., et al. A standardized rat model of volumetric muscle loss injury for the development of tissue engineering therapies. *Biores Open Access* 1, 280, 2012
- [5] Corona B.T., Rouviere C., Hamilton S.L., and Ingalls C.P. Eccentric contractions do not induce rhabdomyolysis in malignant hyperthermia susceptible mice. *J Appl Physiol* 105, 1542, 2008
- [6] Criswell T.L., Corona B.T., Ward C.L., Miller M., Patel M., Wang Z., et al. Compression-induced muscle injury in rats that mimics compartment syndrome in humans. *Am J Pathol* 180, 787, 2012